

# The PAZ Polarimetric Radio Occultation Research Dataset for Scientific Applications

Ramon Padullés[1,2], Estel Cardellach[1,2], Antía Paz[1,2], Santi Oliveras[1,2], Douglas C. Hunt[3], Sergey Sokolovskiy[3], Jan-Peter Weiss[3], Kuo-Nung Wang[4], F. Joe Turk[4], Chi O. Ao[4], and Manuel de la Torre Juárez[4]

[1]Institut de Ciències de l'Espai, Consejo Superior de Investigaciones Científicas (ICE-CSIC), c/Can Margans, S/N, Campus UAB, 08193 Bellaterra (Barcelona), Spain
[2]Institut d'Estudis Espacials de Catalunya (IEEC), Barcelona, Spain
[3]University Corporation for Atmospheric Research (UCAR), Boulder, CO, USA
[4]Jet Propulsion Laboratory (JPL), California Institute of Technology, Pasadena, CA, USA

**Correspondence:** Ramon Padullés (padulles@ice.csic.es)

**Abstract.**

Polarimetric Radio Occultations (PRO) represent an augmentation of the standard Radio Occultation (RO) technique that provides precipitation and clouds vertical information along with the standard thermodynamic products. A combined dataset that contains both the PRO observable and the RO standard retrievals, the *resPrf*, has been developed with the aim to foster

the use of these unique observations and to fully exploit the scientific implication of having information about vertical cloud structures with intrinsically collocated thermodynamic state of the atmosphere. This manuscript describes such dataset and provides detailed information on the processing of the observations. The procedure followed at UCAR to combine both H and V observations to generate the equivalent profiles as in standard RO missions is described in detail, and the obtained refractivity is shown to be of equivalent quality as that from TerraSAR-X. The steps of the processing of the PRO observations are detailed,

derived products such as the top-of-the-signal are described, and validation is provided.

Furthermore, the dataset contains the simulated ray-trajectories for the PRO observation, and co-located information with global satellite-based precipitation products, such as merged rain rate retrievals or passive microwave observations. These co-locations are used for further validation of the PRO observations and they are also provided within the *resPrf* profiles for additional use. It is also shown how accounting for external co-located information can improve significantly the effective PRO

horizontal resolution, tackling one of the challenges of the technique.

## 1 Introduction

The Radio Occultation and Heavy Precipitation (ROHP) experiment aboard PAZ satellite is providing, for the first time, Polarimetric Radio Occultation (PRO) observations (Cardellach et al., 2014). This new observational technique represents an augmentation of the standard RO (e.g. Kursinski et al., 1997; Hajj et al., 2002), that measures the incoming Global Navigation

Satellite System (GNSS) signals with a two-linearly orthogonal (H-horizontal; V-vertical) polarized antennae, instead of a right hand circularly polarized (RHCP) one (which is used in all the rest of current and past standard RO missions). The advantage



of collecting the H and V components of the incoming radio waves is that in addition to the standard thermodynamic products, the phase difference between H and V contains information about the hydrometeors encountered along the ray path. Therefore, joint measurements of precipitation and thermodynamics are obtained.

The PAZ satellite was launched in February 2018 with the first ever payload able to collect PRO, which was activated on May 10th that same year. Roughly one year after the launch, the capability to detect rain was demonstrated (Cardellach et al., 2019). Validation of these observations has been performed extensively: Cardellach et al. (2019) and Padullés et al. (2020) showed the good agreement with two-dimensional precipitation products, such as the NASA's Integrated Multi-satellitE Retrievals for Global Precipitation Mission (GPM, IMERG, Huffman, 2017) and the merged product of infrared brightness temperatures

($T_B$) at 10.8 $\mu$m from geosatationary satellites (Janowiak et al., 2017). In addition to precipitation, sensitivity to the upper layers of convective clouds has been demonstrated (Padullés et al., 2022), and a good climatological agreement with Ice Water Content (IWC) retrievals is observed (Padullés et al., 2023). More recently, precise validation of the vertical structure of the signal has been performed using the US Next Generation Weather Radar (NEXRAD) polarimetric observations (Paz et al., 2024). Therefore, PRO is able to provide a view of the vertical structure of clouds and precipitation complementary to that

from other satellite platforms, along with the intrinsically collocated thermodynamic state of the atmosphere.

    Applications taking advantage of PRO capabilities have been already investigated and proposed as missions (e.g. Cardellach et al., 2017; Turk et al., 2019, 2022), as well as for potential applications for Numerical Weather Prediction (NWP) (e.g. Murphy et al., 2019), including efforts towards an operational forward operator to assimilate PRO observables into NWP models (Hotta et al., 2024). Furthermore, interest from commercial companies in the technique has already materialized, with Spire Global

Inc. launch of three PRO-capable nano-satellites in early 2023.

    Being the ROHP-PAZ experiment a proof-of-concept mission and the first of its kind, the equipment on PAZ was not fully optimized to collect PRO. Besides the fact that the antenna is polarimetric (able to collect H and V independently), no other major changes were made in the receiving system. For example, the two antenna cables coming out from the H and V ports are using the receiver input ports that are usually reserved for the forward and aft antenna connections. Consequently, PAZ does

not carry a forward antenna and is therefore not collecting rising occultations. The most important consequence of this is that the measurements in H and V are not synchronized in time, but there is a slight difference on the tracking start time.

    This manuscript aims at describing a new dataset made available at the Institut de Ciències de l'Espai - Consejo Superior de Investigaciones Científicas (ICE-CSIC), Institut d'Estudis Espacials de Catalunys (IEEC), https://paz.ice.csic.es, which contains both the polarimetric and the standard thermodynamic products to facilitate the scientific exploitation of the observations

(Padullés et al., 2024). The dataset is called *resPrf*, arising from *research profiles* following the well established University Corporation for Atmospheric Research (UCAR) naming for RO missions products. It also contains valuable information on the precipitation context of the observations, that has also been shown important when available (e.g. Turk et al., 2021). Such information is provided already interpolated into the PRO ray-paths, and contains information from IMERG, infrared $T_B$, and from the different channels of the GPM radiometer constellation coincident observations (Kidd et al., 2021). These data

products aim at fostering the scientific use of PRO and to speed up the process towards their operational use.



**Table 1.** Overall organization of the *resPrf* files. A detailed description of each group is provided in AppendixA.

| main group | sub group | inner group | variables |
|---|---|---|---|
| profiles | | | $\Delta\Phi$ profile |
| | | | RO profiles |
| rays | | | latitude |
| | | | longitude |
| | | | height |
| colls | precipitation | | precipitation |
| | IR $T_B$ | | IRTb |
| | PMW | swath | channel $T_B$ |

The *resPrf* files are in *netcdf-4* format (Unidata, 2023) and organized in nested groups, variables and global attributes. Such organization is shown in Table 1. The first group, called *profiles*, contains the vertical profiles of the polarimetric observable $\Delta\Phi$, and of several thermodynamic variables, all interpolated into the same vertical grid from 0 to 40 km altitude every 0.1 km. The second group, called *rays*, contains the ray-traced locations of the signal trajectory between the GPS satellite and PAZ,
re-gridded and sub-sampled so that the tangent point of each ray corresponds to the same vertical grid as for the vertical profiles, up to 20 km (i.e. resulting in 200 rays between 0 and 20 km), plus 20 more rays reaching up to 60 km at lower vertical resolution.

And the third group, called *coll*, contains the collocated data from IMERG precipitation surface rain rates and the Infrared $T_B$ interpolated onto the rays locations. Also, this groups contains collocated passive Microwave (PMW) brightness temperatures
(TB) from each of the PMW radiometers that make up the GPM satellite constellation (Kidd et al., 2021). A more detailed description of all variables is provided in Appendix A.

Besides describing the dataset, this manuscript also tackles a few other important aspects of the PRO data. The first one is to describe the methodology used to retrieve the standard thermodynamic products from the polarimetric observations, a process performed by UCAR. The quality of such retrievals is assessed, since the use of a polarimetric antenna (instead of the RHCP
one, which matches the emitted polarization) may result in a degraded quality of the signal. The second one is to describe the processing of the polarimetric observable and to assess its quality using the collocated precipitation information. Furthermore, derived products from the $\Delta\Phi$ are also described, and a brief discussion about the horizontal resolution of the polarimetric observables is included.

The manuscript is organized as follows: it starts with a brief section highlighting the main aspects of the PRO technique.
Section 3, Section 4, and Section 5 describe the corresponding groups of the *resPrf* dataset, with details on the processing, interpolation and collocations, respectively. Section 6 evaluates the retrievals and obtained products, such as the standard thermodynamic profiles and the $\Delta\Phi$. Further focus on the quality and applications of the $\Delta\Phi$ is provided in the sub-sections within Section 6. Finally, Section 7 shows an illustrative example on how the provided ray trajectories can be used in conjunction with



model outputs to mimic a 2D forward operator and simulate PRO observations. The manuscript ends with the conclusions.
Furthermore, the appendix contains detailed information about the data and metadata of the *resPrf* useful for the users of the dataset.

## 2 PRO technique in a nutshell

The PRO technique was first proposed by Cardellach et al. (2014) as an extension of standard RO. The idea is to collect GNSS occulting (or rising) signals from a LEO using a dual-polarized antenna (horizontal, H, and vertical, V). The signals traveling
from the GNSS satellite to the LEO cross different layers of the atmosphere, and the ray trajectories bend due to vertical refractive gradients as the radio-link penetrates into the Earth's atmosphere (e.g. Kursinski et al., 1997). In the lowermost portion of the occultation event, electromagnetic (EM) signals travel nearly tangential to the surface inside the troposphere, even reaching within the boundary layer (e.g. Sokolovskiy et al., 2006; Ao et al., 2012). Therefore, these rays are affected by precipitation if they encounter rain events along the ray-paths.

The idea of PRO is to compare the phase measured at the H and V ports of the antennae to infer information about precipitation and clouds in addition to the traditional thermodynamic RO products obtained from their bending. Precipitating hydrometeors such as raindrops, snow, or frozen particles in clouds can be asymmetric and are usually horizontally oriented (e.g. Gong and Wu, 2017; Zeng et al., 2019). These induce a phase delay that is larger in the horizontal component than in the vertical, and it can be measured by comparing the phase observations at the H and V ports of the antennae.

The observable representing the differential phase delay experienced by the EM propagating wave per unit length is the specific differential phase shift ($K_{dp}$):

$$K_{dp} = \frac{\lambda^2}{2\pi} \int \Re\{f_{hh} - f_{vv}\} N(D) dD \tag{1}$$

where $D$ is the equivalent particle diameter, $N(D)$ is the particle size distribution, $f_{hh,vv}$ represent the co-polar components of the scattering amplitude matrix, $f$, and $\Re$ indicates its real part. $\lambda$ is the wavelength (in this case, $\sim 0.19$ m corresponding
to L1 frequency band). It is worth noting that in Eq. 1, an extra $\lambda/2\pi$ is included to express $K_{dp}$ in units of mm/km, to be consistent with the GNSS community and previous literature about PRO. Therefore, information about the the distribution of sizes is provided through the $N(D)$, and the information about the asymmetry of the particles, shape, and density is contained in $f$.

The actual observable for PRO is then defined as the total differential phase shift, obtained for each observation time:

$$\Delta\phi(t) = \phi_h(t) - \phi_v(t) = \int_L K_{dp}(l) dl \tag{2}$$

where $\phi_{h,v}$ is the phase of the EM wave measured at each port $h$ or $v$, $L$ is the length along the ray path under the influence of rain or clouds, and $l$ is the ray-path length position. In practice, $\Delta\phi$ can be obtained using the *excess phase* observable ($\Phi$) obtained through the standard RO processing of both H and V signals as if they were independent (see next Section).





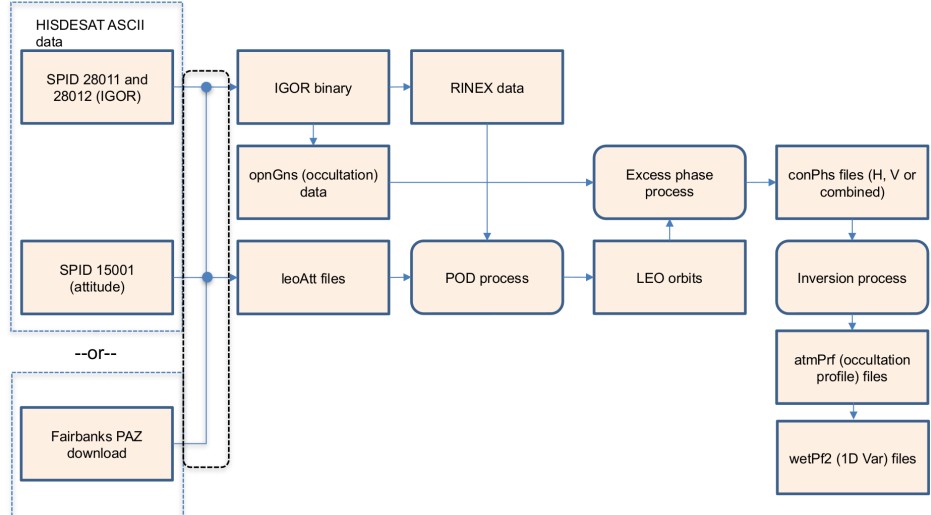

**Figure 1.** PAZ data processing at UCAR/CDAAC. Custom H/V combining software is included in the excess phase process.

## 3 Dataset description: profiles

The first main group of the dataset is the *profiles* group. This contains the vertical profiles of $\Delta\phi$, as well as the standard RO thermodynamic retrievals from UCAR's *wetPf2*. All of them are interpolated into a gridded vertical coordinate between 0 and 40 km, with a resolution of 0.1 km.

### 3.1 Standard RO processing

The PAZ processing chain for standard RO products at the UCAR COSMIC Data Analysis and Archive Center (CDAAC)
using combined H/V polarization is shown in Figure 1. It starts by reading the data from the PAZ satellite, obtained either through Hisdesat (satellite operators) or through the data down-link at Fairbanks. These data come as IGOR binaries and are converted to *opnGns* at the CDAAC.

Standard RO processing is done separately on the H and V polarization signals. The standard RO processing at UCAR/C-DAAC follows these steps.

1. Start with high rate occultation data in the *opnGns* format. *opnGns* is a custom data format defined at UCAR for quick and flexible storage of high rate GNSS data as described in Weiss and Hunt (2022).

2. Remove orbital motion (LEO and GNSS POD data), GNSS clocks (e.g. IGS products), and LEO clocks (via differencing with a high elevation 'reference' satellite) to create an *excess phase* profile.

3. Compute an atmospheric Doppler model from climatology.

4. Integrate this model to get a phase model, then difference it with the excess phase computed above.



5. This phase angle $\Delta\theta$ between successive samples is now rotating slowly enough to generate meaningful I and Q components: $I = SNR \times \cos(\Delta\theta), Q = SNR \times \sin(\Delta\theta)$.

6. Apply GPS navigation bits to the open-loop portion of I and Q.

7. Stitch open- and closed-loop I's and Q's together.

8. Compute phase via $\mathrm{atan2}(Q, I)$.

9. Fix full cycle slips by adding or subtracting $2\pi$ to minimize the difference between samples.

10. Add the phase model back in to get connected excess phase.

11. These connected L1 and L2 phase profiles are then submitted to the *inversion process* to compute bending angle, refractivity, and finally temperature and pressure profiles. This same inversion process is used by all CDAAC missions (Sokolovskiy, 2014).

12. Finally, the dry atmospheric temperature and pressure profiles are combined optimally with ECMWF weather model data in a 1D variational assimilation process (Wee et al., 2022).

The resultant H and V profiles are nearly identical, but the SNR and penetration depth can be enhanced by combining them at the excess phase level via a vector sum of the I and Q data. Here is the procedure:

1. Determine a 'master' polarization for this occultation. We use higher SNR to choose between H and V.

2. Compute separate Horizontal and Vertical I and Q profiles as in the single-polarization procedure outlined above.

3. Determine the phase alignment between H and V. For PAZ, the H and V polarization data use separate antenna channels. Thus H and V phases have slightly different time stamps and can be out of phase alignment.

4. Line the I's and Q's up and use the 'master' to fix $\frac{1}{2}$ cycle slips in the 'slave'.

5. Find the point at which the slave polarization signal descends into noise. This is the point at which to stop the vector combination, and continue with only the master polarization.

6. Perform a vector sum of the lined up I and Q values from the master and slave: $I = I_s + I_m, Q = Q_s + Q_m$.

7. Assemble the combined excess phase (as in steps 9-10 in the single polarization processing).

8. Compute the SNR of the combined signal as $\frac{\sqrt{I^2 + Q^2}}{\sqrt{2}}$. This assumes similar random noise on H and V channels.

The resultant H/V combined profiles have higher SNR than either the H- or V-only profiles and penetrate slightly deeper in the atmosphere. In addition, the count of successful occultations when combining H and V is higher than for H or V only.





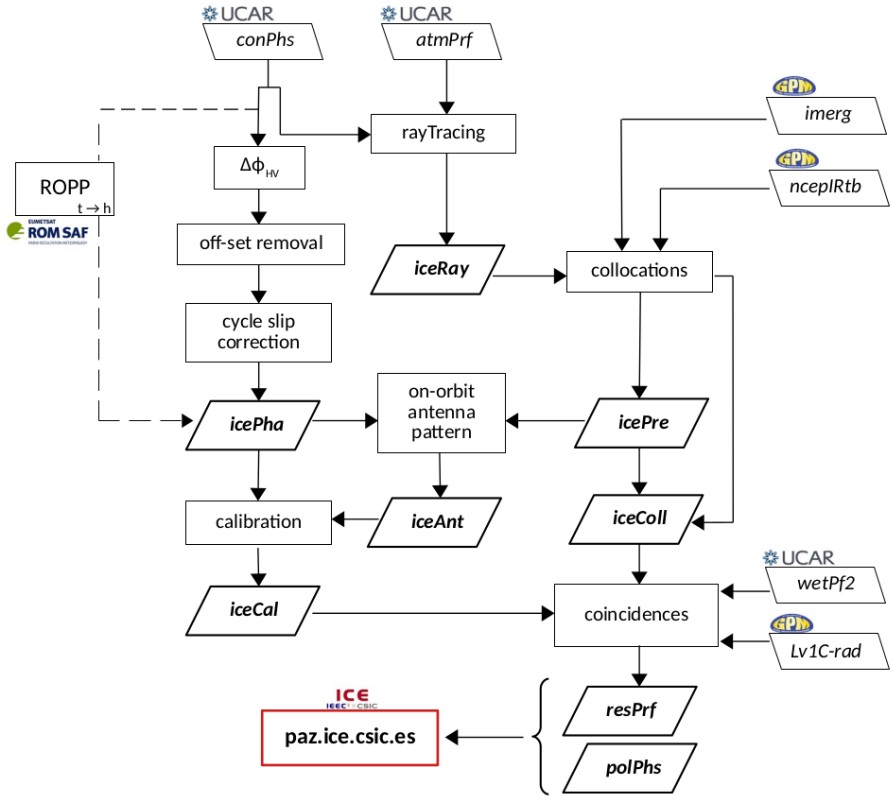

**Figure 2.** Processing chain implemented at ICE-CSIC,IEEC to generate the *resPrf* profiles. Input file-types are respresented as a top-left tilted rectangles, while output files and intermediate files generated at ICE-CSIC,IEEC are represented as top-right tilted bold rectangles. Processing steps are represented as regular rectangles. See text for more details on each of the steps.

## 3.2 Polarimetric processing

The polarimetric processing is conducted at the ICE-CSIC, IEEC. The complete processing chain is shown in Figure 2. The algorithm to process these data that is detailed here expands upon what was described in Padullés et al. (2020).

The processing of the polarimetric part starts with the *conPhs* files provided by UCAR (UCAR-COSMIC-Program, 2023a). These files contain the RO observables as function of time, only that in the PAZ case most variables are provided for each polarization (e.g. $SNR_{H,V}$, excess $phase_{H,V}$, etc.). At this point, the first important step is already done by UCAR, that is, to interpolate both H and V observables to the same time stamp. This enables the straight-forward differentiation of the H and V excess phases to obtain the main PRO observable:

$$\Delta\Phi(t) = \Phi_H(t) - \Phi_V(t) \tag{3}$$





where $\Phi$ is the L1 excess phase in each polarization, and $t$ is time. Even though RHCP should have a $\Delta\Phi = 90°$, since the observables used are the excess phases, the differentiation yields an arbitrary number, that can be set to 0 in the higher layers of the atmosphere, where no precipitation is expected. Therefore, the rest of the $\Delta\Phi(t)$ observation down below is relative to that initial value.

In order to obtain a height ($h$) linked to each time measurement, a modified version of the Radio Occultation Processing Package (ROPP, Culverwell et al., 2015) is used. The link between $t$ and $h$ is based on geometric optics, with all the limitations and consequences it may have (e.g. under strong atmospheric multipath, a large ambiguity is expected in associating a single $h$ to a $t$ measure). Estimations performed at Padullés et al. (2020) predicted an uncertainty of more than 0.5 km below 2 km altitude. Hence, altitude assignment for heights below 2 km are not to be fully trusted. Ideally, $\Delta\Phi$ should be obtained through
wave optics retrievals, and work towards this is being pursued (e.g. Wang, K.-N. et al., 2021).

     After the differentiation of the $\Phi_{H,V}$, jumps in the $\Phi$ compatible with uncorrected cycle slips become noticeable. To correct for those, half cycle correction is applied to the Closed Loop (CL) portion of the profile, and full cycle correction is applied to the Open Loop (OL) portion (Padullés et al., 2020, Sect. 2.1). These corrected profiles are stored in internal (ICE-CSIC,IEEC) intermediate files called *icePha*. Those files obtained under no-rain conditions are used to build the on-orbit antenna pattern,
which is used to calibrate the whole dataset (Padullés et al., 2020, Sect. 5). The precipitation information is stored in the intermediate files called *icePre* and *iceColl*, that are described in Section 5. The calibrated phase is also smoothed using a 1 sec smoothing filter. The final calibrated and smoothed $\Delta\Phi$ is the main PRO observable and is stored in the intermediate files called *iceCal*, and later in the processing copied into the *resPrf*. The $\Delta\Phi$ provided in the *resPrf* is re-gridded so that it corresponds to the same height as the thermodynamic retrievals (from 0 to 40 km each 0.1 km).

**3.2.1 Height flag indication**

An important attribute included in the *resPrf* is the *height_flag*. This is a parameter that aims at indicating under which height the $\Delta\Phi$ is not trusted anymore, in terms of the quality of the $\Delta\Phi$ itself. It identifies jumps in the $\Delta\Phi$ profile that were not corrected during the cycle slip correction, often associated to tracking issues and low SNR, or uncorrected cycle slips that were not properly identified as such.
To perform the identification of altitudes where $\Delta\Phi$ is no longer trustable, three conditions must be met by the data:

1. A first sliding window of 50 points is applied to the corrected 50 Hz $\Delta\Phi$ data, where the standard deviation is computed ($SD_1$). This $SD_1$ must exceed a threshold of 10 mm, which would mean that a jump in the $\Delta\Phi$ is detected. Such jumps may be due to noisy isolated measurements, and $\Delta\Phi$ may recover rapidly. To ensure that is not the case, another condition must be met.

2. Another sliding window of 50 points is applied then to the calibrated and smoothed $\Delta\Phi$ profiles, and the standard deviation is computed ($SD_2$). Here, the threshold that $SD_2$ must exceed is 1.5 mm. This indicates that after smoothing, there is still some remaining issues in $\Delta\Phi$. If condition (1) above were met due to isolated noisy measurements, the smoothing would result in a $SD_2$ lower than the specified threshold.





3. Finally, $SD_2/|\Delta\Phi|$ must exceed an arbitrary threshold of 0.4. This relative magnitude discards increases in the $SD_2$ that are due to a sudden increase of $\Delta\Phi$ caused by rapidly entering into a heavy precipitation area.

If all these three conditions above are met simultaneously, the height at which this happens is stored in the *height_flag* parameter. The profiles are considered to have good quality above that height, but users are advised against using the $\Delta\Phi$ below *height_flag*.

## 4 Dataset description: rays

The RO technique is a limb-sounding measurement. Despite the fact that often RO retrievals are associated to a single point location (known as occultation point), the truth is that a larger portion of the atmosphere is contributing to the bending angle, and therefore to refractivity and derived products. In the case of standard RO, the contribution is maximized around the tangent point, with an effective horizontal resolution of ~100-150 km in the direction along the rays (e.g. Anthes, 2011).

For PRO, the contribution does not maximize anywhere along the ray-paths, and therefore contribution from all areas crossed by the rays must be equally taken into account. That is why an indicative representation of all ray-path locations is very helpful to better interpret and understand the PRO measurements.

The *resPrf* files have two groups storing variables in this regard. The first one is the group *rays*, which contains the locations (latitude, longitude, and height) of the points of the selected rays. The group *colls* contains the external precipitation information interpolated into these ray-point locations. Further details are provided in the subsequent subsections.

### 4.1 Ray-tracing and interpolation

A ray-tracing software is used to simulate the actual ray-path trajectories between the GPS and PAZ satellites at each time stamp of the occultation. To properly account for bending of the rays, the retrieved refractivity profile is used, and therefore ray-path trajectories are only available for those profiles whose corresponding UCAR *atmPrf* profile is available and passed the quality control. Each ray is computed solving the geometrical optics equations accounting for the time-dependent locations of the satellites. Therefore, the resulting collection of rays is a slant plane that realistically represents the tangent point drift, due to the relative movement between the satellites.

Once all ray trajectories are obtained, these are sub-sampled and re-gridded so that the tangent points coincide with the vertical coordinate and resolution of the thermodynamic and $\Delta\Phi$ profiles described above: that is, 200 rays are provided with tangent points between 0 and 20 km, one each 0.1 km. In addition, 20 more rays are included with tangent points reaching up to 60 km, with lower vertical resolution. Each of the 220 rays contain 301 points in the along-ray dimension, implying an along-ray resolution of about 5 km for the portion of the rays below 20 km. The interpolation of all the data available (those further explained in Section 5) is performed using the nearest neighbor method, using the latitude and longitude of the rays and discarding the height dimension (since the interpolated data is 2-Dimensional).

An example of the traced rays can be seen in Figure 3, where the portion of the rays below 20 km is shown projected onto the surface, with different background images in each panel. This case corresponds to a coincidence between PAZ and



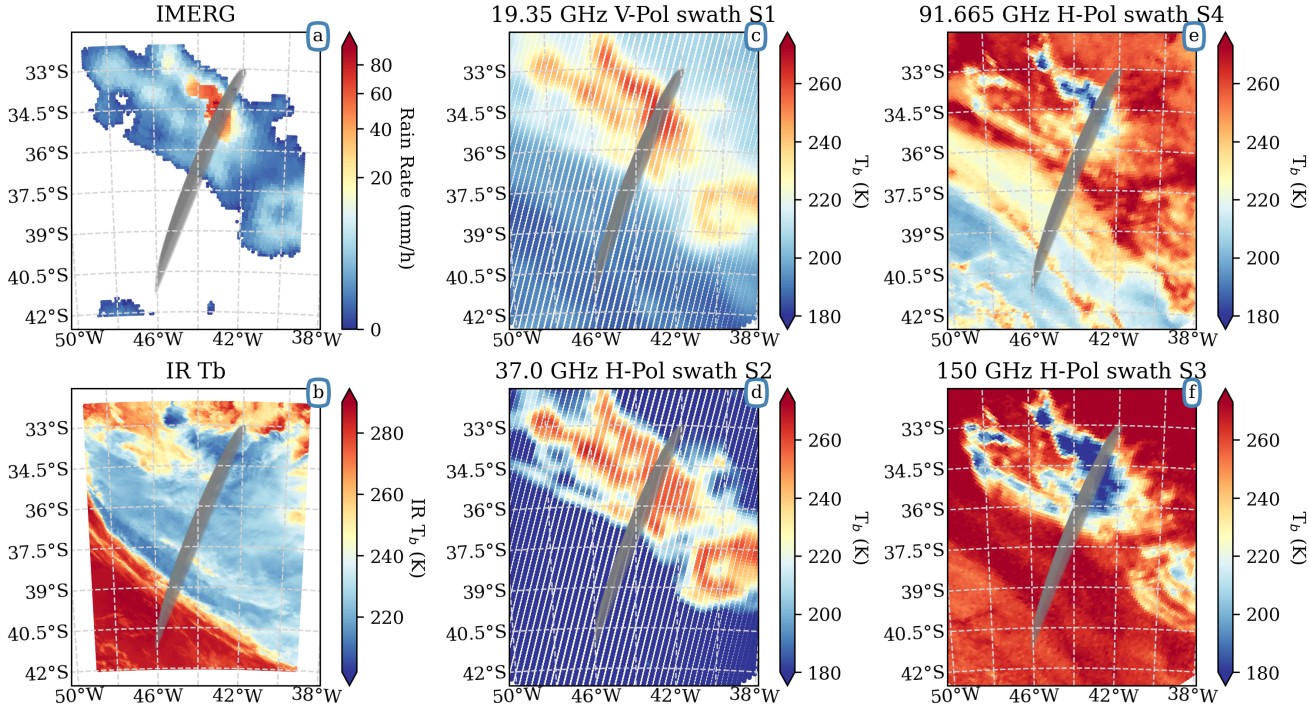

**Figure 3.** Example of a collocation of a PAZ observation with ID PAZ1.2019.126.09.11.G12 with IMERG (panel a), IR $T_B$ (panel b), and different channels of the PMW radiometer SSMIS aboard DMSP F17, as indicated at the top of each panel. The gray zone represent the projection on the surface of the PRO rays portion below 20 km.

the Defense Meteorological Satellite Program (DMSP) F17 satellite, where the IMERG and IR data were also available. The different background images in Figure 3 correspond to surface rain rate, IR $T_B$, and 4 different channels of the DMSP F17 PMW radiometer. More details about these collocations is provided in the following sections.

The user should become familiar with the moon-slice shape of these ray projections. The longer rays correspond to rays with their tangent point at lower altitudes. As the rays' tangent point increases in altitude, a shorter portion of the rays lays below 20 km, therefore, it appears shorter in these projection maps.

## 5 Dataset description: collocations

### 5.1 IMERG

The GPM IMERG precipitation products provide surface rain rates with global coverage, with an horizontal resolution of 0.1° and a time resolution of 30 minutes. Being a two-dimensional product, the interpolation is performed into the projection onto





the surface of the PRO rays, regardless of the height dimension. A limitation is placed based on height: only the portion of the rays below 20 km is used, since it is conservatively assumed that no precipitation reaches those heights.

Despite some known limitations (e.g. Watters et al., 2023; Li et al., 2022; Peinó et al., 2022; Ramadhan et al., 2022), the use of IMERG is beneficial for validating PRO observations thanks to its global coverage and time resolution. The IMERG data interpolated onto the rays is provided in a sub-group called *precipitation*, within the *colls* parent group. The rain rate at each ray-point location is stored in the variable *precipitation*, and the group also has several attributes like the average rain rate around the occultation point.

### 5.2   $Tb_{11}$

Likewise IMERG, the merged IR 10.8 $\mu$m $T_B$ product offers global coverage (+/-60° latitude) and 30 minutes time resolution, with an horizontal resolution of $\sim$4 km. Due to the sensitivity of PRO to frozen hydrometeors and to some extent, to the cloud structure, the use of this product helps identify the portions of the rays within clouds, assuming that IR $T_B$ is informative of the cloud top heights. The IR $T_B$ is interpolated into the rays (also, only to the lat-lon projection of the rays, no height taken into account) and these are provided in a sub-group called *IRTb* within the *colls* parent group. The $T_B$ at the latitude/longitude of each ray point is stored in the variable *IRTb*, which also contains a few attributes providing additional information, like the minimum IR $T_B$ around the occultation point.

Also, this product is used to determine the cloud top height (CTH) of the clouds crossed by the PRO rays. For such determination, the coldest brightness temperature is matched with the retrieved temperature (T) vertical profile from PAZ, and the height at which both coincide is determined. Similarly, assuming that the temperature profile is valid not only at the occultation point but in the whole area sense by the rays, one can determine whether each ray point is below or above the corresponding cloud height by comparing both temperatures. Here, the height dimension of the ray is taken into account, so that the atmospheric temperature profile is linked to each height, regardless of lat/lon. An example of this is shown in Figure 4. Panel (a) shows the IR $T_B$ interpolated into the portion of each of the rays below 20 km. The representation shows the along ray dimension in the x-axis, and the tangent point height linked to each ray in the y-dimension. Then, the atmospheric T from each ray point (only based on height) is compared against the corresponding IR $T_B$ at the same point: if the atmospheric T is colder than IR $T_B$, that point is considered to be above the clouds, i.e. outside the cloud. Otherwise, it is considered to be inside (or below) the cloud. The edge of the rays inside the clouds is shown with a green contour overlayed in Figure 4-(a) and (b). This is further exploited in Section 6.2.2.

### 5.3   Space-based radiometers

Each PAZ observation is checked against overpasses of the GPM-constellation satellites carrying PMW radiometers. The time constrain for the coincidence is set to 30 min, and the distance is constrained by the size of each radiometer' swath: that is, the occultation point of the PAZ observation must be within the swath. The list of satellites with the corresponding sensor, swath widths, number of coincidences with PAZ (until 2023-08-01) and percentage of coincidences within the tropics is provided in Table 2. The number of coincidences scales almost linearly with the time allowed to consider a coincidence, e.g. when reduced


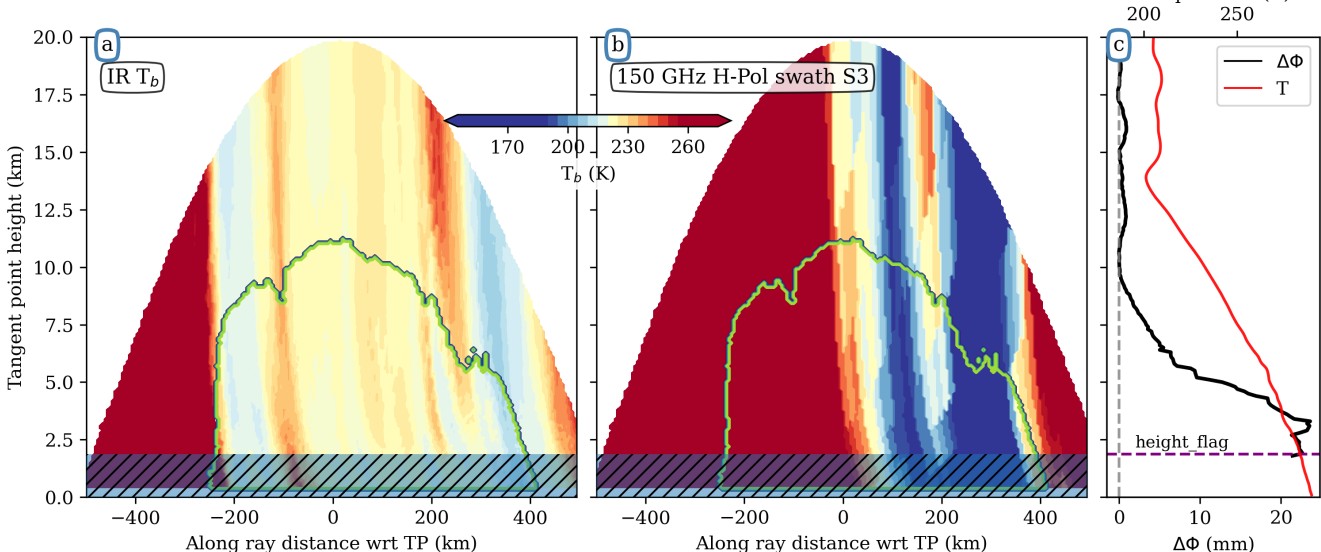

**Figure 4.** Example of the result of the interpolation of the IR $T_B$ (panel a) and the 150 GHz $T_B$ from F17 satellite (panel b) as function of the along ray distance with respect to the tangent point (x-axis) and the tangent point height of each ray (y-axis). This example corresponds to the same case as in Figure 3. The green contour indicate the portion of the rays inside clouds, according to the procedure explained in Section 5.2. Panel c contain the vertical profiles of $\Delta\Phi$ (black, bottom x-axis) and atmospheric temperature (red, top x-axis) as a function of height (shared axis with tangent point height from panels a and b). The hatched area in panels (a) and (b) and the purple dashed line in panel (c) indicate the height of the *height_flag* parameter.

to 15 min, the number of coincidences drops by a 50-55%. An example of a coincidence with the DMSP F17 satellite is shown in Figure 3, where 4 PMW channels are selected (e.g. 19.35 GHz, 37.0 GHz, 91.665 GHz and 150 GHz).

For the cases that have a collocation with one of these PMW radiometers, the $T_B$ corresponding to all channels of the radiometer are interpolated into the PRO rays. The interpolation results for the 150 GHz channel of the case example illustrated in Figure 3 is shown in Figure 4-(b). These data are provided in sub-groups named after the collocated satellite (e.g. *F17*, *GPM-GMI*, *NOAA-19*, etc.) within the *colls* parent group. Each sub-group contains an inner group corresponding to the swath mode, and each of these swath-related groups contain the data for each radiometer channel in the corresponding variables. Frequency and polarization of each variable are provided in the corresponding attributes.

## 6   Evaluation of retrievals

The dataset contains the standard RO retrievals and the calibrated $\Delta\Phi$. The following sub-sections are dedicated to the validation of these products.





**Table 2.** List of satellites with passive microwave radiometers that have coincidences in space and time with PAZ observations, from 2018-05-10 until 2023-08-01. The number of coincidences and the % of them in the tropics is determined by the orbit of the corresponding satellite and its orbit local time of the ascending node. Note that PAZ is in a 0930/1830 orbit. More detailed information can be obtained in Turk et al. (2021). The acronyms read as follow: GPM: Global Precipitation Measurement; GCOM-W1: Global Change Observing Mission for Water; NPP: National Polar Orbiting Partnership; GMI: GPM Microwave Imager; SSMIS: Special Sensor Microwave Imager/-Sounder; AMSR: Advanced Microwave Scanning Radiometer; ATMS:Advanced Technology Microwave Sounder; MHS: Microwave Humidity Sounder; SAPHIR: Sounder for Probing Vertical Profiles of Humidity.

| satellite | sensor | swath width (km) | coincidences | 30S-30N (%) |
|---|---|---|---|---|
| GPM | GMI | 880 | 10364 | 24 |
| F16 | SSMIS | 1700 | 31333 | 9 |
| F17 | SSMIS | 1700 | 68813 | 37 |
| F18 | SSMIS | 1700 | 45329 | 18 |
| GCOM-W1 | AMSR2 | 1450 | 10147 | 0 |
| NPP | ATMS | 2500 | 16636 | 0 |
| NOAA-19 | MHS | 2500 | 53010 | 21 |
| NOAA-20 | ATMS | 2500 | 16634 | 0 |
| METOP A | MHS | 2500 | 17443 | 0 |
| METOP B | MHS | 2500 | 16185 | 0 |
| METOP C | MHS | 2500 | 10741 | 0 |
| MEGHA-TROPIQUES | SAPHIR | 1700 | 9353 | 100 |

## 6.1  Standard thermodynamic products

One of the challenges of the PRO technique is its ability to provide good quality thermodynamic (or standard) retrievals with an antenna performing with a -3 dB loss. This is due to the fact that each linear polarization captures half of the power due to the polarization mismatch (with respect to a circularly polarized antenna). This could have implications for the penetration depth and quality of retrievals specially in the lower troposphere. To quantify this effect, comparisons have been performed with TerraSAR-X (hereafter, TSX) RO observations. TSX is a rather similar satellite platform carrying a standard RO antenna and the same RO receiver as PAZ (i.e. an IGOR+, a modified version of the one used by the COSMIC-1 satellites, manufactured by Broad Reach Engineering), orbiting at the same height and same orbit inclination of 97.4° (Beyerle et al., 2011). The antenna in PAZ is similar to the one in TSX, both manufactured by Haigh–Farr, and only modified in PAZ to capture H and V polarizations instead of RHCP.

Therefore, comparisons of PAZ PRO observations with TSX RO provide a suitable benchmark to quantify the compromise that PRO undergoes for providing precipitation. The data used for the comparison are the *atmPrf* retrieved by UCAR and the corresponding collocated *echPrf* for the comparison with ECMWF model (UCAR-COSMIC-Program, 2023b).





**Figure 5.** Penetration depth and O-B comparison between TSX and PAZ refractivity retrievals, for all data between 2019-01-01 and 2019-06-30. Panel (a) corresponds to the percentage of total profiles (y-axis) reaching a certain penetration height (x-axis), for all retrieved data. Panel (b) shows the same as panel (a), but for observations obtained in the Tropics (25S-25N) and panel (c) shows the corresponding results for extratropical cases (25S-90S, 25N-90N). Panel (d) corresponds to the fractional differences (x-axis) as a function of height (y-axis), for all retrieved data. Panel (e) shows the same as panel (d), but for observations obtained in the Tropics (25S-25N) and panel (f) shows the corresponding results for extratropical cases (25S-90S, 25N-90N). The shaded area represent the 10th to 90th percentile of the distribution of fractional differences as function of height.

The first thing being compared is the penetration depth. The results for the minimum height at which refractivity is being retrieved for both PAZ and TSX at UCAR CDAAC is shown in Figure 5-(a-c). It shows the result of the percentage of profiles reaching a certain height for all PAZ and TSX observations during the first half of 2020, separated by tropical and extratropical observations. It can be seen how the penetration depth is slightly better for TSX, a difference that is larger in the tropics. In the





tropics, below 2 km, the difference between TSX and PAZ is of around 2.5%, and below 1 km is around 4% increasing with decreasing altitude. Outside the tropics the difference is negligible.

Regarding the quality of the profiles, the comparison is performed using the ECMWF model background as reference. All
300 TSX and PAZ refractivity profiles obtained during the first half of 2020 are compared with the refractivity obtained from the collocated ECMWF model (*echPrf*). The fractional refractivity difference, i.e. $(N_{obs}-N_{ecmwf})/N_{obs}$, as a function of height is shown in Figure 5-(d-f). No significant differences can be observed in the comparison, indicating that the quality of the retrievals is equivalent.

## 6.2 Polarimetric products: Differential Phase Shift

The way used to evaluate $\Delta\Phi$ is to group these profiles based on the precipitation under which they were obtained, and the results are shown in Figure 6. The first thing to look at is the group with no precipitation. Such group contains those profiles where the collocated IMERG precipitation is $P_{IMERG} = 0$ mm/h, and should have a mean $\Delta\Phi(h)$ equal to 0 mm for the whole vertical profile. This is shown in Figure 6-(right) with the solid black line. It can be seen how it is following very closely the 0 mm line. Also, Figure 6-(left) shows the vertical profile of the standard deviation ($SD_{\Delta\Phi}$) of the group corresponding
to $P_{IMERG} = 0$ mm/h. It can be seen how the standard deviation is very small (e.g. $SD_{\Delta\Phi}<0.3$ mm), and it increases with decreasing altitude. It reaches $\sim$1.5 mm at an approximate altitude of 2 km, which agrees very well with previous studies (e.g. Cardellach et al., 2014; Padullés et al., 2020). When precipitation increases, the corresponding group mean $\Delta\Phi$ increases as well. This also corroborates previous studies (e.g. Cardellach et al., 2019; Padullés et al., 2020), and implies that $\Delta\Phi$ is sensitive to precipitation intensity. Note that $P_{IMERG}$ used here represents an average of all rain rates interpolated onto the rays
(accounting only for the portion of the rays below 6 km). Since this may cover a relatively large area, values of the order of 2 mm/h are already representative of substantial precipitation.

It is also worthwhile noting the shape of the profiles obtained under heavy precipitation (i.e. roughly when $P_{IMERG} > 2$ mm/h). At this point it is noticeable how the maximum of the $\Delta\Phi$ is achieved at altitudes around 5 km or higher. Even though here it is shown with the mean $\Delta\Phi$ profiles in the corresponding precipitation groups, this feature is clear in individual
profiles obtained under heavy precipitation. There are two main reasons behind this shape: (1) The first one is that the PRO observable $\Delta\Phi$ is sensitive to frozen hydrometeors, as long as they are horizontally oriented (e.g. Padullés et al., 2022, 2023), and these are placed above the freezing level; (2) The second reason is purely geometrical: the PRO rays with tangent points around 5/6 km are traveling a long distance within clouds (rather spatially homogeneous), whereas the lower rays are crossing more portion of precipitation, that is in turn more spatially inhomogeneous. Therefore, contribution to $\Delta\Phi$ maximizes at these
heights.

At the time of writing, the main hypothesis is that oriented snow is the major contributor to the $\Delta\Phi$ observable (e.g. Padullés et al., 2022; Hotta et al., 2024). Besides the scattering properties of such hydrometeors, their contribution is also favored by the fact that the uncertainty of the PRO measurement arount $\sim$4-6 km is already very low, therefore much better than below 3 km where the effect of liquid precipitation maximizes. Furthermore, at the lowest heights there could be a certain ambiguity

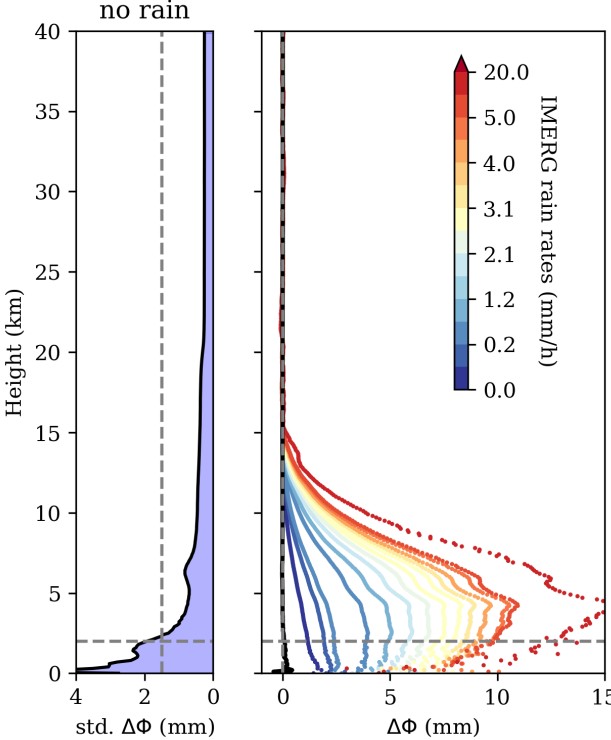

**Figure 6.** Statistics for the $\Delta\Phi$ profiles. Left panel shows the standard deviation as function of height of the group of $\Delta\Phi$ obtained when $P_{\mathrm{IMERG}} = 0$ mm/h, representing the free-of-rain cases. Right panel shows the average as function of height of the $\Delta\Phi$ grouped by $P_{\mathrm{IMERG}}$, when exceeding a certain value indicated in the colorbar. The black line is the average for $P_{\mathrm{IMERG}} = 0$ mm/h.

coming from atmospheric multipath, due to the use of geometric optics to obtain the link between time and height (Padullés et al., 2020).

### 6.2.1 Polarimetric products: Top of the Signal

The fact that $\Delta\Phi$ is sensitive to frozen hydrometeors, and that these are horizontally oriented, globally, and for different types of clouds (e.g. Gong and Wu, 2017; Zeng et al., 2019; Padullés et al., 2023) implies that the $\Delta\Phi$ signature should have a 335 relationship with the tops of the clouds. Being a measurement at L-band, it is expected that sensitivity to clouds starts below the cloud top height (as inferred from the infrared, for example). Therefore, here the relationship between the top of the $\Delta\Phi$ signal and the CTH around the observation (obtained using the IR 10.8 $\mu$m $T_B$) is investigated.

First of all, the CTH is derived by matching the minimum IR 10.8 $\mu$m $T_B$ around the occultation point (inside a circle of $2°$ of diameter) with the retrieved vertical temperature profile. The height at which the temperatures match is identified as the 340 IR-derived CTH. This value is stored in the *cth_irT2deg* parameter. This methodology has some caveats, such as that it can fail





for thin clouds, multilayered clouds, etc. However, it provides a rough estimation of the CTH that is reliable specially in the tropics.

The top of the signal (TOS) is derived solely using the vertical $\Delta\Phi$ profile. First, a mean ($m_{\Delta\Phi}$) and standard deviation ($SD_{\Delta\Phi}$) of the free-of-clouds portion of the profile, i.e. above potential clouds, is obtained. It is computed between 18 and 345 30 km. Then, a threshold is defined as $\mathrm{thresh} = m_{\Delta\Phi} + 3 \times SD_{\Delta\Phi}$. Once this threshold is obtained, $\Delta\Phi$ values exceeding $\mathrm{thresh}$ are identified. Starting from the top, the first 5 consecutive measurements where $\Delta\Phi(h) > \mathrm{thresh}$ are identified. TOS is then defined at the height of the first of such five consecutive measurements. The height at which this happens is stored in the *deltaphi_top_height* parameter. If the condition is not met, *deltaphi_top_height* is set to 0.1 km.

Obtained TOS are compared with IR-derived CTH for those cases where TOS>0.1 km, and the results are shown in Figure 7-350 (a) using a density plot. In addition, the average TOS per CTH bin and the associated standard deviation are provided as well. It can be seen how even though there exist a robust relationship showing an increase in TOS as the IR-derived CTH increases, there also exists a region (i.e. large CTH with low TOS) that separates the relationship from the perfect match. This issue is linked mainly to a geometry problem: the IR-derived CTH may be far from the tangent point.

To account for the geometry, the IR-derived CTH is now substituted by the highest point of the PRO rays that are considered 355 to be inside the cloud, as explained in Section 5.2. This is an equivalent procedure that accounts for the actual ray trajectories and not only for the immediate environment. The comparison of the TOS with the IR-derived CTH when the geometry is accounted for is shown in Figure 7-(b). Now the relationship is much closer to the perfect 1:1 match than the results shown in panel (a).

This results have two important implications. The first one is that the agreement between the TOS and IR-derived CTH 360 demonstrates the sensitivity of the PRO $\Delta\Phi$ to clouds. The relationship between TOS and CTH is linear, with a bias of around $\sim$2 km, meaning that the sensitivity is larger at the infrared than at L-band, something totally expected. The second implication of the results in Figure 7 is that it is very important to account for the actual trajectories and the context to interpret the PRO $\Delta\Phi$ observable. This is something similarly stated by Hotta et al. (2024), where the authors determine that the use of a 2D forward operator is crutial, and a 1D forward operator would fail to properly use $\Delta\Phi$ for NWP experiments.

### 6.2.2 Polarimetric products: Horizontal Resolution

The main challenge of the PRO $\Delta\Phi$ observable is its horizontal resolution. Unlike the standard RO level-2 products, where the contribution maximizes around the tangent point, in the PRO case, the hydrometeors contribute equally all along the ray-paths. Therefore, the horizontal resolution is the length the rays travel below the height at which one expect to find clouds. Table 3 shows the effective horizontal resolution of rays with different tangent point heights depending at which height one assumes 370 the CTH is.

An interesting feature is that the horizontal resolution depends on height, and therefore different height points on the $\Delta\Phi(h)$ profile have different effective horizontal resolution. In fact, the resolution improves with increasing altitude and becomes really small when approaching the CTH. Based on the results in the previous Section 6.2.1, instead of the CTH, the TOS could be used to determine the height at which one starts to expect clouds. Another important characteristic is that the effective

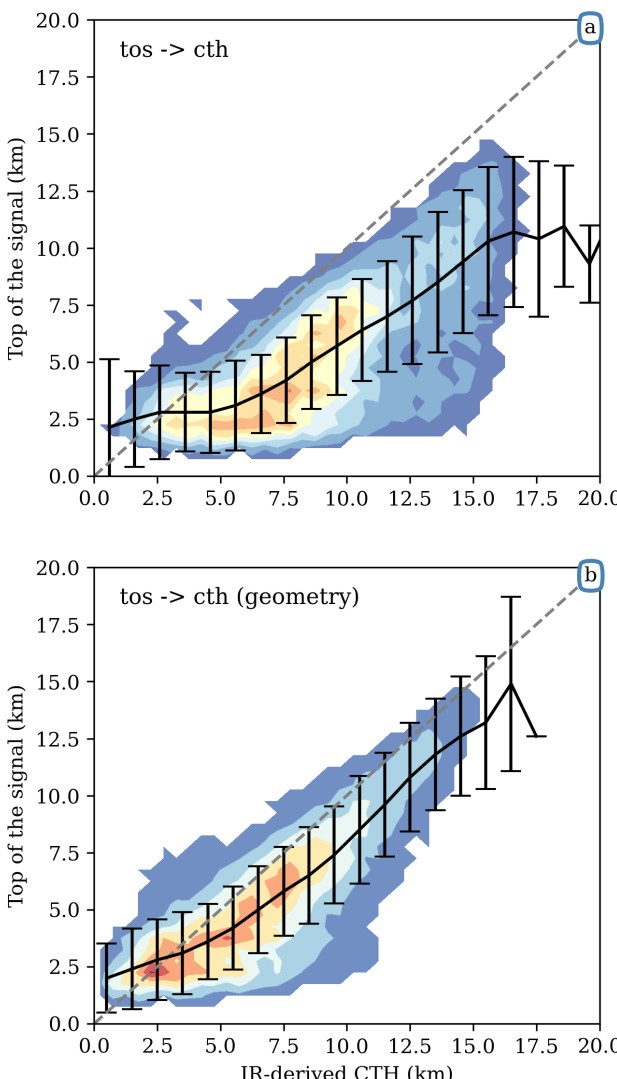

**Figure 7.** Density plot (cold colors less, warm colors more data) comparing the $\Delta\Phi$ top of the signal (TOS) (y-axis) and the IR-derived cloud top height (CTH) (x-axis), when the minimum IR $T_B$ around the occultation point is used (panel a) and when the geometry of the actual ray trajectories are taken into account (panel b). Black line represents the average and the error bars the standard deviation of the TOS within each 1-km CTH bin.

resolution determined by the length traveled by the rays below a certain height is only providing an upper limit to the resolution, but the actual horizontal resolution is likely less than that.

In the cases where precipitation information like what is described in Section 5 is available, the actual horizontal resolution can be determined using the interpolated data. For example, in the case illustrated in Figure 4, one can see that the CTH accounting for the geometry of the rays is approximately 11 km. This would imply an upper limit on the horizontal resolution



**Table 3.** Effective horizontal resolution (in km) of a ray with a certain tangent point height depending on where CTH is assumed.

| TP height | CTH | | | |
|---|---|---|---|---|
| | 18 km | 16 km | 14 km | 12 km |
| 2 km | 962.8 | 902.7 | 832.6 | 762.5 |
| 5 km | 852.3 | 782.2 | 707.0 | 626.9 |
| 10 km | 656.4 | 566.3 | 461.1 | 325.8 |
| 12 km | 566.1 | 460.9 | 325.7 | $\sim 0$ |

of almost 600 km for a ray with a tangent point of 5 km. However, using the portion of the rays that are within the cloud (according to the interpolated IR $T_B$ and the retrieved temperature profile, see Section 5.2), it can be determined that the actual horizontal resolution is ~540 km, as it can be inferred from the green contour in Figure 4-(a).

This methodology could be pushed further by using the interpolated PMW data and setting a constrain in the $T_B$ to identify portions of the cloud that not only the ray is crossing, but that are actually contributing to $\Delta\Phi$. Only as an illustrative example,
in the case of Figure 4, one could assume that the only portion of the cloud that contributes to $\Delta\Phi$ is the one where $T_B$ from the 150 GHz channel is below 220 K. Then, the horizontal resolution of the ray with tangent point height of 5 km would drop to ~300 km, as it can be inferred from Figure 4-(b).

## 7 Illustrative example: col-location with ERA-5

This Section aims at illustrating the potential use of the ray-path trajectories that are provided in the *resPrf* files. With them, it
is straightforward to know the actual locations of the rays and to be used as a input for a 2D forward operator, in combination with model outputs with the aim to simulate $\Delta\Phi$ profiles.

Figure 8 shows an example of utilization of the ray-path trajectories. For this illustrative case, the corresponding ERA5 reanalysis (Hersbach et al., 2020, 2023) fields are used. The fields of interest to simulate $\Delta\Phi$ are the ones containing information about hydrometeors, that is, the mass of water, snow, cloud ice, and cloud liquid particles. These quantities are obtained in the
fields *specific rain water content*, *specific snow water content*, *specific cloud ice water content*, and *specific cloud liquid water content*, respectively (Hersbach et al., 2023). Must be noted that these hydrometeor fields are obtained from the large-scale clouds generated by the cloud scheme in the ECMWF Integrated Forecasting System (IFS).

The specific water contents are converted into water content (WC) densities. These are obtained at each point of the model grid, and then interpolated into all the ray trajectories crossing that region (which are provided in the *resPrf* files). WC are
converted to $K_{dp}$ following the same approach as in Padullés et al. (2022) and Hotta et al. (2024), that is, using a simple approximation that depends on the effective density ($\rho$) of the hydrometeor, axis ratio ($ar$), and the amount of hydrometeors, e.g:

$$K_{dp}^i \propto WC_i \times \rho_i \times (1 - ar_i) \tag{4}$$

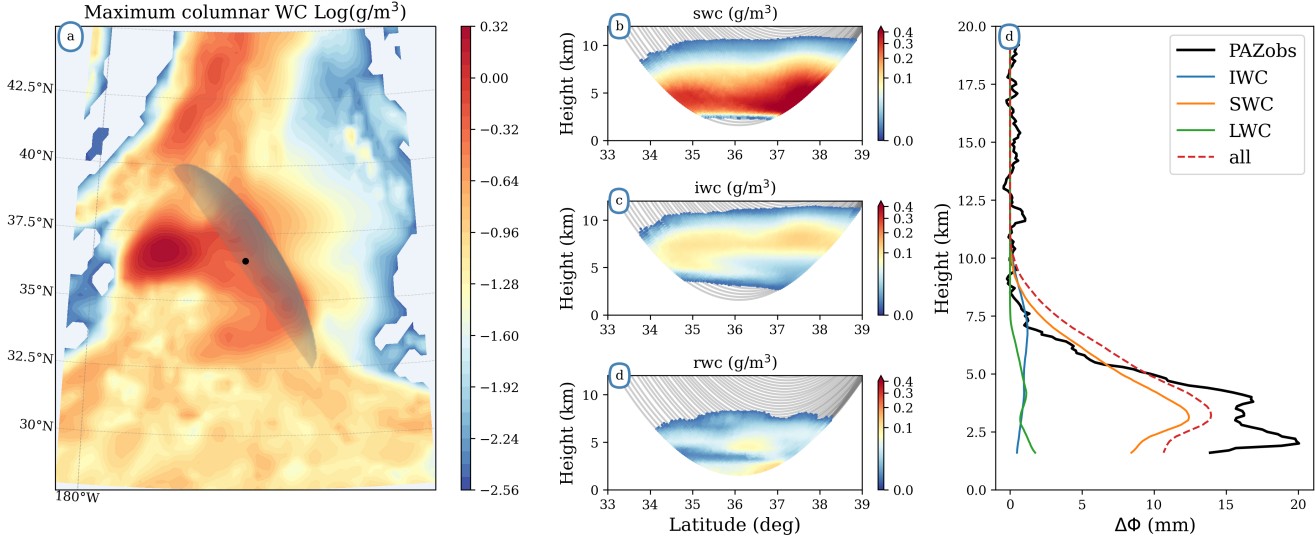

**Figure 8.** Collocation of PAZ observation with ID PAZ1.2020.355.16.46.G19 and the corresponding ERA5 water content fields. Panel a shows the projection of the PRO rays trajectories (only portion of the rays below 20 km) on the surface, with color background indicating the maximum columnar water content from ERA5. Panels b, c and d show the WC fields interpolated into the ray-trajectories provided by te *resPrf* files accounting for the 3D information, for snow, ice and rain, respectively. Panel e shows the $\Delta\Phi^i$ for all considered hydrometeors, along the total sum for all hydrometeors (dashed) and the actual PAZ observation (black).

where $i$ denotes each of the hydrometeors in consideration. For each ray, $K_{dp}^i$ is integrated along the whole ray-path to obtain $\Delta\Phi^i$.

This is exemplified in Figure 8. The maximum WC in the whole surrounding area of the PRO event is shown in panel-a, while the interpolation of the WC corresponding to different hydrometeors into the ray-trajectores is shown in the central panels (b,c, and d). The result of the integration, therefore $\Delta\Phi^i$, is shown in the panel-e, in comparison with the PAZ $\Delta\Phi$ observation. For this case, a fixed $\rho$ of 200 kg/m$^3$ is being used for the frozen hydrometeors, and an axis ratio of 0.5, 0.8, and 0.7 is being used for snow, ice, and rain, respectively. The results show a good agreement between the PAZ $\Delta\Phi$ and the simulated $\Delta\Phi$, and emphasizes the contribution of snow.

More importantly, this example shows a straightforward way to use freely available data (e.g. *resPrf* and ERA5) to perform comparative analyses of $\Delta\Phi$, with potential to be used in assessments of model (e.g. operational models from different agencies, NWP models, etc.) performance, microphysics evaluation, scattering parametrization, among others.

# 8 Conclusions

The *resPrf* dataset is created and publicly disseminated with the aim to foster the scientific use of the PRO observations. These provide unique observations of the vertical structure of clouds and precipitation with intrinsically collocated thermodynamic





soundings of the atmosphere. Since 2018, the vertical profiles of $\Delta\Phi$ have been publicly provided in the *polPhs* files. Now, these and the future PAZ GNSS PRO observations have been reprocessed and their output files expanded: the new dataset contains

also the specific locations describing the full sounded slant plane, with external data interpolated into it. These external data consists of rain rate retrievals from the IMERG, IR $T_B$ observations, and radiances from PMW radiometers. These data are all provided through netcdf-4 files organized in nested groups, variables and the corresponding attributes, making the data rather self-explanatory.

The procedure to combine the H and V observations from PRO to obtained standard RO products is described in detail in

Sect. 3.1. The description provides all steps so that the methodology can be reproduced and applied to other PRO missions. The obtained products have been shown to have equivalent quality as the retrievals from the TerraSAR-X mission, a very similar satellite platform collecting standard RO. However, in the results shown here, quality of PAZ profiles is slightly lower in the tropics. Very recently, Spire Global has shown results of a similar approach (i.e. comparing standard RO retrievals with those obtained from PRO observations after combining H and V), and the quality of the combined H and V retrievals is even better

than their regular standard RO retrievals (e.g. presentations in Turk et al., 2024). This implies that the results from PAZ may be affected from the fact that the PRO equipment was not fully optimized, and the fact that there are artificial artifacts affecting the antenna surroundings (Padullés et al., 2020). Also, the Spire Global antennae designs, being different from that in PAZ, may play a role. Nevertheless, both sets of results encourage the use of PRO, provided that in addition of equivalent-like standard thermodynamic products one also obtains the precipitation information.

The description of the polarimetric processing chain is provided in Sect. 3.2. Furthermore, a detailed methodology to detect spurious portions of the $\Delta\Phi$ vertical profile is provided, and labeled as *height_flag*. The resulting $\Delta\Phi$ profiles are validated using collocations with IMERG, showing results that agree with previous validation studies and that further indicate the sensitivity of PRO to precipitation intensity and to oriented frozen hydrometeors.

All the collocations from IMERG, IR $T_B$, and PMW radiometers provide additional information on the context of the

observations. In addition, it emphasizes the complementarity of the limb-sounding like observations from PRO, providing additional insight in the vertical perspective on 2D observations. This may be of special importance in complementing the observations of PMW radiometers with polarized measurements (i.e. GMI, F17, etc.) at high frequencies (e.g. > 90GHz), with vertical information on oriented hydrometeors inducing strong signals. At the same time, these collocations provide additional validation to the PRO products, as it has been shown with the $\Delta\Phi$ top of the signal comparison with the CTH derived from IR

$T_B$.

It has been also shown how collocated data can help tackle the issue of horizontal resolution in PRO. Theoretically, the horizontal resolution of $\Delta\Phi$ can be as large as the ray length, constrained within the portion traveling under the potential height of cloud tops. Using external information such as the IR $T_B$, or the PMW radiances, the effective horizontal resolution can be further constrained. Although using external information is not always desirable, products like the merged IR $T_B$ are

(almost) always available within $\pm60°$ latitude, and can be used routinely to constrain the upper limit of the $\Delta\Phi$ horizontal resolution. Horizontal resolution can represent an issue for scientific applications, and therefore this kinds of solutions are





**Table A1.** Description of the global attributes of the files.

| global attribute | description |
|---:|:---|
| roid | PAZ occultation event identifier as in UCAR CDAAC |
| timeUTC | Start time of the occultation event in UTC |
| lon_occ | Longitude of the occultation point, from UCAR atmPrf |
| lat_occ | Latitude of the occultation point, from UCAR atmPrf |
| az_surf | Azimuth: angle with respect to north of the link between the GPS and PAZ, from UCAR atmPrf |
| ocean | 1: the occultation point is over ocean; 0: occultation point is over land |
| terrain_height | Height above MSL in km of the terrain below occultation point |

worth being explored. However, in the operational use of these observations, since the use of a 2D Forward Operator stands as a requirement (Hotta et al., 2024), the horizontal resolution becomes less of an issue.

This fact is illustrated in Figure 8. Besides the actual comparison between the PAZ observation and the ERA5-derived $\Delta\Phi$ that has been shown, this example wants to highlight how easy is to use the ray-trajectories contained in the *resPrf* files to perform simulations of $\Delta\Phi$. This could be of help to users not having a 2D forward operator available to perform assimilation studies and model performance assessments.

Overall, the presented dataset should encourage the community to continue exploring these observations, given the potential they have shown.

**Appendix A**

This Appendix provides a set of tables describing in detail the different groups withing the *resPrf* data files. Table A1 describes the global attributes of the files. Table A2 describes the variables and attributes of the netcdf group *profiles*. Table A3 provides the description of the variables in the group *rays*. Finally, the table A4 provides a description of the sob-groups and inner groups within the main group *colls*, along with their corresponding variables and attributes.

**Appendix B**

This appendix contains the reference DOI for each PMW satellite product that are used in this work.

**Data availability**

The described datasets *resPrf* are available at https://paz.ice.csic.es/ (DOI 10.20350/digitalCSIC/16137) (Padullés et al., 2024). The corresponding *atmPrf* and *wetPf2* profiles are available at https://data.cosmic.ucar.edu/gnss-ro/paz/postProc/ (DOI:10.5065/k9vg-



**Table A2.** Description of the profiles group.

| variable | dimension | unit | description |
|---|---|---|---|
| height | (400) | km | Height above the Mean Sea Level (MSL) |
| dph_smooth | (400) | mm | Smoothed differential phase shift into 0.1 km vertical grid |
| dph_smooth_std | (400) | mm | Standard deviation of the smoothed differential phase shift into 0.1 km vertical grid |
| temperature | (400) | K | Wet temperature from UCAR wetPf2 interpolated into 0.1 km vertical grid |
| vp | (400) | mb | Water vapor pressure from UCAR wetPf2 interpolated into 0.1 km vertical grid |
| pressure | (400) | mb | Pressure from UCAR wetPf2 interpolated into 0.1 km vertical grid |
| sph | (400) | g/kg | Specific humidity from UCAR wetPf2 interpolated into 0.1 km vertical grid |
| rh | (400) | % | Relative humidity from UCAR wetPf2 interpolated into 0.1 km vertical grid |
| gph | (400) | km | Geopotential height from UCAR wetPf2 interpolated into 0.1 km vertical grid |
| refractivity | (400) | N | Refractivity from UCAR atmPrf interpolated into 0.1 km vertical grid |

| group attribute | | | description |
|---|---|---|---|
| height_flag | | | Height flag parameter as described in Section 3.2.1 |
| deltaphi_10km | | | Mean of $\Delta\Phi$ between minimum valid height (indicated in height_flag) and 10 km. |
| deltaphi_15km | | | Mean of $\Delta\Phi$ between minimum valid height (indicated in height_flag) and 15 km. |
| deltaphi_max | | | Maximum value of $\Delta\Phi$. |
| deltaphi_max_height | | | Height at which deltaphi_max is encountered. |
| deltaphi_top_height | | | Height at which the top of the signal is found. |
| deltaphi_top_height_tresh | | | Threshold value $\Delta\Phi$ needs to exceed to be selected as TOS. |
| deltaphi_rms20 | | | Root Mean Square (RMS) of $\Delta\Phi$ between 20 km and 40 km. |

**Table A3.** Description of the rays group.

| variable | dimension | unit | description |
|---|---|---|---|
| latitude | (220,301) | deg | Latitude of the ray-point, for 220 rays each defined in 301 points |
| longitude | (220,301) | deg | Longitude of the ray-point, for 220 rays each defined in 301 points |
| height | (220,301) | km | Height above the MSL of the ray-point, for 220 rays each defined in 301 points |



**Table A4.** Description of the colls group.

sub-group: precipitation

| variable | dimension | unit | description |
| --- | --- | --- | --- |
| precipitation | (220,301) | mm/h | IMERG surface precipitation rain rates interpolated at the (lon,lat) coordinates of each ray point |

| group attribute | | | description |
| --- | --- | --- | --- |
| filenameImerg | | | Name of the IMERG source file |
| meanPrecip_2deg | | | Mean precipitation within a circle of 2° diameter around the occultation point. |
| meanPrecip_06deg | | | Mean precipitation within a circle of 0.6° diameter around the occultation point. |

sub-group: IRtb

| variable | dimension | unit | description |
| --- | --- | --- | --- |
| IRtb | (220,301) | K | IR $T_B$ interpolated at the (lon,lat) coordinates of each ray point |

| group attribute | | | description |
| --- | --- | --- | --- |
| filenameIR | | | Name of the IR $T_B$ source file |
| irTemp_2deg | | | Minimum IR $T_B$ within a circle of 2° diameter around the occultation point. |
| cth_irT2deg | | | Cloud Top Height obtained by matching irTemp_2deg with the atmospheric temperature vertical profile. |

sub-group: satellite name

inner-group: swath

| variable | dimension | unit | description |
| --- | --- | --- | --- |
| channel $T_B$ index | (220,301) | K | PMW $T_B$ corresponding to the channel indicated in the variable attributes interpolated at the (lon,lat) coordinates of each ray point |

| group attribute | | | description |
| --- | --- | --- | --- |
| satellite | | | Name of the satellite |
| sensor | | | Name of the PMW radiometer sensor aboard that satellite |
| satellite_obstime | | | UTC time of the closest observation to the occultation time |
| timediff | | | Difference in minutes between PMW observation and occultation time |
| dx | | | Distance in km between the nadir track of the satellite and the occultation point |





**Table B1.** DOIs for the GPM constellation products.

| satellite | sensor | DOI |
|---|---|---|
| GPM | GMI | 10.5067/GPM/GMI/GPM/1C/07 |
| F16 | SSMIS | 10.5067/GPM/SSMIS/F16/1C/07 |
| F17 | SSMIS | 10.5067/GPM/SSMIS/F17/1C/07 |
| F18 | SSMIS | 10.5067/GPM/SSMIS/F18/1C/07 |
| GCOM-W1 | AMSR2 | 10.5067/GPM/AMSR2/GCOMW1/1C/07 |
| NPP | ATMS | 10.5067/GPM/ATMS/NPP/1C/07 |
| NOAA-19 | MHS | 10.5067/GPM/MHS/NOAA19/1C/07 |
| NOAA-20 | ATMS | 10.5067/GPM/ATMS/NOAA20/1C/07 |
| METOP A | MHS | 10.5067/GPM/MHS/METOPA/1C/07 |
| METOP B | MHS | 10.5067/GPM/MHS/METOPB/1C/07 |
| METOP C | MHS | 10.5067/GPM/MHS/METOPC/1C/07 |
| MEGHA-TROPIQUES | SAPHIR | 10.5067/GPM/SAPHIR/MT1/1C/07 |

t494). The TerraSAR-X data used in Sect. 6.1 are available at https://data.cosmic.ucar.edu/gnss-ro/tsx/postProc/ (DOI: 10.5065/fv7s-ax27). The IMERG dataset (DOI:10.5067/GPM/IMERG/3B-HH/07), the merged IR $T_B$ products (DOI:10.5067/P4HZB9N27EKU) and the PMW files (see DOIs in Table B1) are available at the NASA GES DISC (https://disc.gsfc.nasa.gov/).

*Author contributions.* RP and EC led the study. EC is the PI of the ROHP-PAZ experiment. RP, EC, AP and SO have participated in the data processing at the ICE-CSIC, IEEC, and ultimately generated the resPrf products. DH, SS, and JPW developed the processing of the standard
RO products for PRO at UCAR. KNW, FJT, COA, MT participated in the processing, testing, comparison and collocations at the NASA/JPL. All authors contributed to writing and reviewing the manuscript.

*Competing interests.* The authors declare no competing interests

*Acknowledgements.* This publication is part of the Grants RYC2021-033309-I and PID2021-1264436OB-C22 funded by the MCIN/AEI (10.13039/501100011033) and the European Union «NextGenerationEU»/PRTR and "ERDF A way of making Europe". Work performed
at the ICE-CSIC was also partially supported by the program Unidad de Excelencia María de Maeztu CEX2020-001058-M. Part of the investigations at ICE-CSIC,IEEC are done under the EUMETSAT ROM SAF CDOP4. The work by authors KNW, FJT, COA and MTJ was carried out at the Jet Propulsion Laboratory, California Institute of Technology under a contract with the National Aeronautics and Space Administration.





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
