# Peer review of "The PAZ Polarimetric Radio Occultation Research Dataset for Scientific Applications"

_Earth System Science Data, 2024_

## Author Response (AR1)

**Review comments for "the PAZ polarimetric radio occultation research dataset for scientific applications" by Padulles et al.**

First of all, we would like to thank the reviewers for their time and dedication on reviewing this paper. We appreciate the positive comments and we are sure these help improve the manuscript.

Below we provide a point by point answer to all comments, and we attach at the end a document highlighting the differences with respect to the previously submitted version.

This paper summarizes the PAZ satellite core Level 2 product dataset and its usability for scientific applications. The original scientific goal for the PAZ mission was to demonstrate polarimetric RO can be used to retrieve heavy precipitation. Research over the years went beyond the original goal to discover the signal association to clouds, and the dataset described in this manuscript is carefully crafted to be particularly suitable for studying cloud-precipitation-thermodynamic interactions. The dataset contains three main groups: (1) RO and phase difference profiles; (2) rays for limb sounding that accounts for earth's rotation as well; (3) collocated cloud and precipitation products from other satellites, including the IMERG precipitation rate, 10.8 um infrared band brightness temperature (TB) and passive microwave (PMW) TBs. The second group is particularly important for fair study the cloud-precipitation-thermodynamic interactions by synergizing limb and nadir soundings together. Several examples are provided in the manuscript to showcase how this dataset could be useful and providing high-quality measurements.

This is a high quality dataset, not only the data quality, but more importantly the scientific quality. Being able to sensing thermodynamic structures within clouds and precipitation, this dataset contains high potential to help tackling some of the long-standing challenges that the majority of current spaceborne passive remote sensing products having in cloudy and precipitating scenes. Even better, the producers of this dataset (i.e., key authors on this paper), took careful consideration of caveats matching limb measurements with nadir measurements by providing the ray paths and matched TBs along the rays. Hence I strongly support the publication of this dataset to the public.

There are some minor issues that I think should be addressed before the acceptance though, mainly to clear out some parts that can cause confusions. Also, as a standard procedure to publish a dataset, the independent validation part feels a bit too weak in the current version. It would be really helpful to have some additional water vapor and temperature sounding comparison results to show in the revised version against other available RO products (global mean as well as regional error distributions) to assess the overall quality of PAZ's Prf products.

Minor comments:

L124: what's a Doppler model and why you need it for forward modeling (?) or retrieval purpose?

This model is constructed using the LEO and GPS satellite positions and velocities and a climate model. It gives the expected change in signal phase due to the neutral atmosphere. Differencing the excess phase from this model yields a signal that is changing slowly enough to perceive and correct GPS navigation bit flips. This is a relevant step in the low-level processing chain. It is probably too technical for end-users of the data, but we prefer to leave it in the manuscript for completeness. A sentence has been added to clarify a bit this issue: "*This model is constructed using the LEO and GPS satellite positions and velocities and a climate model. It gives the expected change in signal phase due to the neutral atmosphere*".

General procedures from Step 1 to Step 12 around Line 130: there's no discussion regarding the U and V components when you do the decomposition. Are they negligible or your receiver can always perfectly align with the I & Q component? This is never clear to me. Also, it is never mentioned in this manuscript how you tease out possible orientation rotation due to magnetic field in Earth's upper atmosphere when the RO rays pathed through?

It must be pointed out that each polarized component (H and V) is a complex signal with separate I and Q components. There is an alignment required between H and V components in the Paz receiver (since each is tracked on a separate receiver channel). This is touched on in steps 3 and 4 in the next procedure, around lines 143-145. To do this alignment, we look at the beginning of the occultation where signal strength is largest. We note the minimum angle needed to pull the 'slave' signal (H or V) into alignment with the 'master' signal and then apply this offset to the slave signal to maximize SNR before combining signals. Note that this gets rid of any phase difference ($\Delta\phi$) between H and V. This is only done for normal occultation processing with combined H and V, not delta phi processing.

Regarding the magnetic field effect on $\Delta\phi$, this was assessed in Padulles et al. 2020, and an explicit reference to this has been included in the text. In line 180-181, we have added the sentence: "This calibration should also remove any effect induced by the ionosphere into $\Delta\phi$ (Padulles et al. 2020, Sect. 5)"

Fig. 2: It's not clear if resPrf depends on input from UCAR wetPrf2? Does resPrf contain temperature and water vapor retrievals or just the bending angle (or refractivity)?

Yes, resPrf contains the wetPf2 retrievals of pressure, temperature, water vapor pressure, specific humidity, relative humidity, geopotential height and refractivity directly from wetPf2. We have made this more explicit in line 59.

Line 195: how do you tell if a suddent delta_Phi increase is not due to intersecting heavy precip, but rather untrustworthy measurements?

The fact that the SD is relative to the absolute value of $\Delta\phi$, both evaluated in moving windows of 50 points in an already smoothed variables, limits the total value when $\Delta\phi$ remains high due to precipitation. When untrustworthy measurements occur, $\Delta\phi$ tends to go back to small values, and then the total value used here (i.e. $SD_2/\Delta\phi$) increases. The value set as a threshold (i.e. 0.4) is admittedly arbitrary, and has been chosen after examining statistics and individual profiles.

Section 4.1, paragraph 1: this paragraph is really confusing. Up till reading this paragraph that I started to realize that your resPrf is different from the standard atmPrf and wetPrf products that UCAR produce for the PAZ mission. Could you provide a percentage rate how many PAZ RO measurements pass the QC and can successfully generate atmPrf and wetPrf at UCAR?

The resPrf is different from the regular UCAR retrievals. It only contains the thermodynamic information about pressure, temperature, water vapor pressure, specific humidity, relative humidity, geopotential height and refractivity directly from wetPf2. We include the following sentence upfront in the introduction (line 59) to avoid confusion:

*"The information contained in the thermodynamic variables comes from UCAR's wetPf2 retrieval products, disseminated here along with the polarimetric polarimetric observables for user convenience."*

Line 240: this is another major part that confuses me: so does the "coll" group contain the imagery from collocated nadir sensor measurements (PMW TB, IR TB, IMERG) or the sensor TBs are instead projected to the ray paths as shown in Fig. 4?

All observables and measurements from IR TB, IMERG, and PMW TB are interpolated into the ray paths. Since these are 2D products, these are interpolated into the projections on the surface of these ray paths. This is explained in the new version of the manuscript, lines: 243, 254, 279.

Line 260: one caveat for this assumption is that there's no temperature inversion layer and no large perturbations (e.g., caused by gravity wave near the convective clouds) that cause artifacts in your decision process, correct? Please clarify here.

The inversion layer in accounted for. However, large perturbations are not. This is clarified in the text now, line 347: *"This methodology has some caveats, such as that it can fail for thin clouds or multilayered clouds, large temperature perturbations (like those induced by gravity waves), etc."*

Line 271: does your "coll" product also include the view-angle from cross-track scanning PMW measurements? TB needs to be corrected for the slantwise view-angle or otherwise the climatology can be nontrivially biased.

This is a very important point, and was not included in the current version of the data. Thank you for noticing it. This will therefore be included in the dataset in the next re-processing, as a new variable within the inner-group "swath", corresponding to each PMW satellite sensor. A sentence has been added to warn about this issue:
*"The view-angle from cross-track scanning PMW instruments has not been taken into account in this version of the data, and this information is going to be included in the next re-processing of the data."*

Fig. 5: caption misses to describe what is N and what is Nmodel. Reading the context around Fig. 5 making me wonder about the temperature and water vapor retrieval qualities and whether that's part of this dataset that needs to be validated or assessed at least.

Caption of Fig. 5 has been updated explicitly mentioning that N is the refractivity and Nmodel is refractivity from  ECMWF.

Regarding the quality of wet retrievals:

This dataset (resPrf) provides the information directly from wetPf2 profiles (generated by UCAR) interpolated into the same vertical levels as $\Delta\phi$. This part of the resPrf is just a "re-dissemination" of the thermodynamic products generated by UCAR, for convenience of the users, that have all information (polarimetric and non-polarimetric) within the same data product. The wet retrievals from UCAR are obtained using a 1DVar technique from retrieved refractivity and model background a priory, and these products have been extensively validated in the literature and are widely accepted within the community as one of the products of reference. Validation of wet retrievals is non-trivial, since depend on assumptions, processing centers, retrieval techniques, used background, against what are the validations performed, etc., and we believe that is beyond the scope of this work.

This work has assessed refractivity as a first retrieval, with the main aim of showing that its general quality is equivalent to that from other similar standard RO missions. However, it is also beyond the scope of this work to assess the quality of the retrieval itself, which has been assessed by other authors extensively, e.g. :

Wee, T.-K., et al., Atmospheric GNSS RO 1D-Var in Use at UCAR: Description and Validation. Remote Sens. 2022, 14, 5614. https://doi.org/10.3390/rs14215614

Shao, X., et al., Consistency and Stability of SNPP ATMS Microwave Observations and COSMIC-2 Radio Occultation over Oceans. Remote Sens. 2021, 13, 3754. https://doi.org/10.3390/rs13183754

Fig. 6: caption misses to explain the grey dashed lines.

Caption of Fig.6 has been updated.

Section 6.2.1: you misses a very important reason to explain the underestimation of CTH from your method: for cloud/snow ice, delta_phi is only large when particles are dominantly horizontally oriented, which not happens in the convective core but in anvils (which is below the convective core) or stratiform precipitating region. Please add in this explanation as well if seeing appropriate.

This is also a very good point, thank you for the suggestion. It has been included in the discussion as a possible explanation to the underestimation.

Fig. 7: change "geometry" to "after geometry correction".

Changed.

Line 396: how do you deal with the cloud fraction (CF) in a grid box?

It is assumed that the sum of all hydrometeor's water content (i.e. cloud ice, cloud liquid, rain, snow) provides the average value for a grid box – according to ERA5 documentation. This is stated in lines 408-409. Furthermore, it is noted in the text that the hydrometeor water content associated with cumulus parametrization is not provided. This is stated in lines 424-426.

Line 414: I think it would better to have some discussions regarding data assimilation of the Delta_phi to plant a seed for future applications, especially considering that your group had done some preliminary work regarding this aspect to make PAZ measurements more useful for NWP.

Thank you for this comment. We have modified the paragraph to give more details about these potential applications, including the use for data assimilation and other NWP related applications. It now reads:

*"More importantly, this example shows a straightforward way to use freely available data (e.g. resPrf and ERA5) to perform comparative analyses of $\Delta\Phi$ , with model outputs. The ray trajectories contained in resPrf and the relationships between Kdp and WC (those provided here and within the aforementioned references, or more complex ones currently being investigated) represent a basic 2D forward operator. This aims to contribute to the development of PRO data assimilation strategies into NWP systems. Similarly, PRO characteristics make this measurement technique potentially useful to*

*assess model performance, such as microphysics schemes discrimination or scattering parametrization evaluation, among others."*

**Review of the manuscript essd-2024-150 titled "the PAZ polarimetric radio occultation research dataset for scientific applications" by Padulles et al.**

First of all, we would like to thank the reviewers for their time and dedication on reviewing this paper. We appreciate the positive comments and we are sure these help improve the manuscript.

Below we provide a point by point answer to all comments, and we attach at the end a document highlighting the differences with respect to the previously submitted version.

The manuscript is a description paper of a new data set of polarimetric radio occultation (PRO) observations produced by PAZ satellite.

PRO observations, like the regular RO bending angle observations, contain information on the atmosphere along the ray path that connects the transmitter GNSS satellite and the receiver LEO satellite. However, PRO observations differ significantly from the regular RO observations in that, unlike the regular RO measurements like bending angles, local spherical symmetry approximation is not justifiable for hydrometeor distributions that are sensed by PRO measurements. To analyze PRO data it is thus primordial to have accurate 3D information of the ray trajectories, but such information is only available after performing ray tracing, which is difficult and hindered the use of PRO observations by wider community.

The new data set presented in this paper is ground breaking in resolving this situation by providing pre-computed ray trajectories along with the PRO measurements. This is an important contribution that is expected to foster and facilitate the use of PRO data from wider users. The paper is also very well organized and well written. I have only minor suggestions that the authors can choose to incorporate or not at their discretion.

Minor comments:

- line 156 " only that in the PAZ case...": maybe you meant to write "the only difference bing that ..."

Thanks for the suggestion. We have rewritten the sentence.

- line 158 and elsewhere: The use of the word "differentitian" to mean the subtraction of V-pol excess phase from H-pol excess phase is confusing. Please consider using different wording, like "differencing" for example.

Changed. Thanks.

- Paragraph starting at line 165: It would be useful to include some short explanation about how geometric and wave optics are different here.

The paragraph has been rephrased to include a brief explanation about geometric and wave optics:

*In order to obtain a height h linked to each time measurement, a modified version of the Radio Occultation Processing Package (Culverwell2015) is used. The link between t and h is based on geometric optics, with all the limitations and consequences it may have: e.g. under strong atmospheric multipath, a large ambiguity is expected in associating a single h to a t measure, since different rays may arrive at the receiver at the same time with different excess Doppler values. Estimations*

*performed at Padulles2020 predicted an uncertainty of more than 0.5~km below 2~km altitude. Hence, altitude assignment for heights below 2~km are not to be fully trusted. Ideally, Δϕ(t) should be obtained through wave optics retrievals, and work towards this is being pursued (e.g. Wang2021). Wave optics aims at disentangling the ambiguity of multi-valued time series using e.g. radio-holographic techniques, that yield one-to-one relationships between excess Doppler and impact parameter under the assumption of spherical symmetry.*

- The first paragraph of section 4.1: I assume, from the description in this paragraph, that the 1D (vertical) refractivity profile from the UCAR retrieval is used to compute the ray tracing assuming that the profile is horizontally uniform. If so, making this point explicit in the text and, if possible, discussing any limitation from this approach would be useful for the data users.

We have included the following sentence: *However, since the refractivity profile is 1D, the effects of horizontal inhomogeneities are not taken into account.*

- line 219 "...one each 0.1km": Put "apart" after 0.1km.

Thanks for the suggestion. Changed.

- line 244 "Likewise": Replace with "Like" or "As with".

Thanks for the suggestion. Changed.

- line 254 "sense": Replace with "sensed"

Corrected.

- line 284, 341 and elsewhere, "specially": Replace with "especially"  (or rephrase).

Thanks for the suggestion. Corrected.

- line 307 "equal to" : consider replacing this with "close to" because, even when there is no precipitation at the ground, there can be hydrometeors aloft.

Thanks for the suggestion. Changed.

- line 328 "arount" : replace with "around"

Corrected.

- Figure 6: Please give precise definition of the "Height". I assume it is the tangent height which is also the lowest height of the ray. If so, please explicitly state so.

We have explicitly stated this in the caption.

- Paragraph starting at line 343: In this paragraph the mean and standard deviation of delta Phi is used, but the statistics is taken over which samples are not clearly stated. Please clarify.

This is computed in the portion between 18 and 30km height. Clarified.

- line 406 "maximum WC": not clear it is maximum with respect to what. I guess it is the largest values of WC over the vertical direction within each column?

Clarified.

- Section 7: An important caveat in interpreting the results shown is section is that, in ERA5, hydrometeor water content associated with cumulus parametrization is not provided. As a result, the KDP equivalent computed solely from large-scale clouds will inevitably underestimate the observed KDP (and hence delta Phi). This will justify the model's underestimation shown in Figure 8d.

This is noted and emphasized in the text.